# Increasing Temperature Activates TREK Potassium Currents in Vagal Afferent Neurons from the Nodose Ganglion

**DOI:** 10.3390/ijms26189119

**Published:** 2025-09-18

**Authors:** Lola Rueda-Ruzafa, Ana Campos-Ríos, Paula Rivas-Ramírez, Manuela Rodríguez-Castañeda, Salvador Herrera-Pérez, José Antonio Lamas

**Affiliations:** 1Department of Functional Biology and Health Sciences, University of Vigo, 36310 Vigo, Spain; lolarrzg@gmail.com (L.R.-R.); privas@uvigo.gal (P.R.-R.); manuela.rodriguez@uvigo.gal (M.R.-C.);salva.herrera@me.com (S.H.-P.); 2Neuroscience Research Group, Galicia Sur Health Research Institute (IIS Galicia Sur), SERGAS-UVIGO, 36310 Vigo, Spain; 3CINBIO, University of Vigo, 36310 Vigo, Spain

**Keywords:** thermal homeostasis, temperature, internal temperature, nodose ganglion, vagus nerve, K2P channels, TREK channels, potassium channels, neuronal excitability, patch-clamp

## Abstract

Temperature homeostasis is controlled by the vagus nerve. Thermal information is collected by thermoreceptors present in the viscera and driven across sensory neurons of the nodose ganglia (NG), which in turn send it to the hypothalamus. While transient receptor potential channels (TRPs) are traditionally considered for thermal transduction, TREK channels belonging to the two-pore domain K^+^ channels family are emerging as thermosensors, but their role in the NG remains understudied. Patch-clamp recordings revealed that increasing the temperature to physiological levels causes a hyperpolarization of the membrane potential followed by a depolarization and, despite physiological temperature increased the firing rate, we have demonstrated that TREK channels might be taking part in the excitability control by counteracting TRPs’ effects. In fact, single-channel experiments revealed an increase in TREK channel open probability and a subsequent rise in their activity in NG neurons. All this indicates that TREK channels, mainly TREK1, may be responsible along with TRPs for the maintenance of the membrane potential at physiological temperature in NG neurons.

## 1. Introduction

In mammals, body temperature maintenance is a highly complex process in which both conscious and unconscious mechanisms participate to ensure adaptation to variable environmental conditions. To achieve thermal homeostasis, these mechanisms require interactions between neural, hormonal and metabolic processes. Most of the intentional behavior animals exhibit to adapt to environmental temperature variations use thermic information captured by neurons in the dorsal root and trigeminal ganglion (DRG and TG, respectively) at skin level [1]. This information travels along the spinal cord to finally reach the somatosensory cortex. This mechanism has been extensively studied [2], and transient receptor potential (TRP) channels seem to play a major role in the detection of thermal stimuli [3,4].

A less well-known but equally important mechanism for survival is triggered by thermal information originating from internal organs. This mainly unconscious information is detected by splanchnic and vagal sensory afferents and is transmitted to the preoptic area (POA), the central thermoregulatory brain region [5]. Vagal sensory afferent cell bodies are in the nodose ganglia (NG), which send information to the nucleus of the solitary tract. NG neurons [6] are classified into A-type neurons (large and myelinated), Ah-type neurons (with intermediate characteristics), and C-type neurons (which are small and lacking in myelination), each playing distinct roles in the transmission of sensory signals [6,7]. They all receive signals from a broad range of visceral organs [8] and play an essential role in the transmission of sensory information [8,9].

Although TRPs are expressed in these ganglia and participate in cold- and warm-temperature transduction [3,10,11], it has been shown that certain nodose neurons can detect changes in temperature even in the absence of TRPs [3], suggesting the involvement of an alternative mechanism. In fact, the depolarization induced by cooling in some DRG neurons is not mainly attributed to TRPs but to the inhibition of a background potassium current [12]. This current, initially called I*_cold_*, closely resembles the current generated by TREK-type two pore domain potassium (K2P) channels. Heat-induced TREK1, TREK2, and TRAAK channels activation has been studied in several models [13,14], including DRG neurons and single C-fibers [15]. To control the effect provoked by TRPs’ activation, TREK channels are activated and counteract the TRP-induced depolarization. This role has been proposed as a protective function for thermosensitivity, since TREK channels’ open probability enhances with the increment of temperature [16,17]. Although still poorly investigated, NG neurons have been reported to express all three TREK subunits [13], but the effect of temperature on NG TREK channels has not yet been investigated. All this information prompted us to study the effects of raising the temperature to physiological levels on NG neurons and the role of TREK channels in regulating excitability.

## 2. Results

### 2.1. Classifying NG Neurons

NG contains the sensitive neuronal somas of vagal afferents [13,18]. Three NG neuronal types, each with characteristic electrophysiological and pharmacological profiles, have been previously identified in both mouse and rat models [6,10]. Accordingly, all neurons recorded in this study were classified prior to temperature experiments to reveal differences in thermal response among A, Ah, and C-type neurons. In this work, a total of 105 (32.82%) neurons were not sensitive to capsaicin (CAP) and did not exhibit TTX-resistant Na^+^ currents, corresponding to A-type neurons (Appendix A). A total of 116 neurons (36.25%) were unresponsive to CAP but displayed both TTX-sensitive and resistant Na^+^ currents, corresponding to Ah neurons (Appendix A). Finally, 99 neurons (30.93%) showed CAP−activated currents and both TTX-sensitive and -resistant Na^+^ currents, corresponding to C-type neurons (Appendix A).

### 2.2. The Increase in Temperature Changes the Membrane Potential

We first aimed to investigate how NG neurons respond when the febrile temperature is exceeded. First, the perforated whole-cell technique in the current-clamp configuration was conducted to assess the effect of increasing temperature on the membrane potential in 25 neurons. Temperature ramps were applied, starting from room temperature (RT) values of 24 °C to temperatures close to 50 °C.

The results showed hyperpolarization followed by depolarization in all three neuronal types (Figure 1A–C). As TREK channels are key controllers of the resting potential maintenance, we wanted to verify whether they might be contributing to the observed heat-induced hyperpolarization. To achieve this, the TREK channel blockers fluoxetine [19] (Figure 1D–F) and spadin [20,21] (Figure 1G–I) were applied to the bath solution prior to increasing the temperature in separate sets of experiments. Applying 100 µM of fluoxetine provoked a significant reduction in heat-induced hyperpolarization across all neuronal types, as did 1 µM spadin, a specific blocker of TREK1 [20,21] (the most abundantly expressed TREK channel in NG [13]), which significantly reduced heat-induced hyperpolarization in A-, Ah-, and C-type neurons (Figure 1J–L, one-way ANOVA with Tukey’s test for multiple comparisons, *p* < 0.01 for A-type neurons and *p* < 0.001 for Ah- and C-type neurons). It is noteworthy that no differences were observed between fluoxetine and spadin applications. Hyperpolarization values measured at 37 °C and depolarization values measured at 50 °C, along with the number of cells recorded for each neuronal type, are detailed in Table 1. Depolarization data are shown in Appendix A. No significant differences were found in the heat-induced maximal hyperpolarization among the three neuronal groups, nor for the depolarization (one-way ANOVA, *p* > 0.05). Data details are included in Table 1.

### 2.3. Physiological Temperature Increases Neuronal Excitability

Before and after raising the temperature, depolarizing current pulses from −50 to 350 pA (50 mV increments, 1 s duration each) were injected to study the effect of a temperature increase on NG neuronal excitability. In another set of experiments, neurons were incubated with 1 µM spadin before increasing the temperature to assess TREK channels’ involvement in excitability. Surprisingly, despite expecting a firing reduction because of the previously documented heat-induced hyperpolarization, raising the temperature from 24 °C to 37 °C resulted in a significant increase in action potential (AP) firing across all three neuronal subtypes (Figure 2, two-way ANOVA with Tukey’s test multiple comparisons, * *p* < 0.05 and ** *p* < 0.01), and neurons showed an even greater increase in their firing rate in the presence of 1 µM spadin (Figure 2, two-way ANOVA with Tukey’s test multiple comparisons, # *p* < 0.05 and # *p* < 0.01). No differences were assessed between AP firing at 24 °C and at 24 °C in the presence of spadin. All this suggests that the depolarization following hyperpolarization might be responsible for increasing the neuronal firing, even if TREK channels are actively taking part in moving the membrane potential to more negative values. Population data are detailed in Table 2.

Additionally, individual AP parameters were measured at 24 and 37 °C (see the Section 4) to study the differences between them, as observed in representative traces from NG neuronal subtypes (Figure 3A–C), and phase plots were constructed to study changes in firing velocity (Figure 3D–F). Latency (Figure 3G), amplitude (Figure 3H), duration (Figure 3I), AHP (Figure 3J), and frequency (Figure 3K) were compared between 24 and 37 °C, showing significant differences, except for duration, which was only decreased in A-type neurons (paired *t*-test, *p* < 0.05, *p* < 0.01 and *p* < 0.001). To investigate changes in velocity, phase plots were constructed, and the threshold (Figure 3L), maximum rate of depolarization (VDmax, Figure 3M), and maximum rate of repolarization (VRmax, Figure 3N) were analyzed, revealing significant differences in all three neuronal types (paired *t*-test, *p* < 0.05, *p* < 0.01 and *p* < 0.001). Detailed populational data are shown in Table 3.

Taken together, these measurements indicate that neurons not only fire more APs but that they do so at a faster rate. Despite a hyperpolarization of the membrane potential is observed when rising temperatures (Figure 1) and could be related to a delay in the initiation of AP firing, the following depolarization might participate in increasing the firing rate of NG neurons. Since heat-sensitive TRPs can generate depolarizing currents upon activation [17,22,23], and we have seen that TREK channels are active at 37 °C (Figure 1 and Figure 2), it is plausible that a dynamic interplay occurs between the hyperpolarizing currents, presumably driven through TREK channels, and the depolarizing effect of TRPs, ultimately enhancing neuronal excitability. Activation of TRPs may thus counteract TREK-induced hyperpolarization, contributing to the increased AP firing rate observed in Figure 2.

### 2.4. Heat Induces an Increase in Macroscopic TREK-like Current and Conductance

To study the specific participation of TREK channels in membrane potential changes in relation to temperature, mouse NG neurons were voltage-clamped at −30 mV to increase TREK current open probability [14] and a cocktail of ion channel blockers (cocktail A) was added to the bath solution in order to block voltage-dependent sodium, potassium, cationic, and calcium currents and pharmacologically isolate TREK currents (see the Section 4). As expected from current-clamp gap-free experiments (Figure 1), the temperature increase induced a clear outward current in all the tested neurons. Importantly, this heat-induced outward current was strongly inhibited by 100 µM of fluoxetine in A-type (Figure 4A, *n* = 8, paired *t*-test), Ah-type (Figure 4B, *n* = 11, paired *t*-test), and C-type neurons (Figure 4C, *n* = 6, paired *t*-test).

Similarly, in a separate set of experiments, neurons were voltage-clamped at −30 mV and 15 mV hyperpolarizing negative voltage pulses were applied every 50 ms (0.5 Hz) to assess changes in membrane conductance. Reversible temperature ramps were carried out in the presence of cocktail A in A-type (*n* = 5), Ah-type (*n* = 9) and C-type neurons (*n* = 10), where temperature was first increased to physiological levels and consecutively decreased back to RT, revealing that currents returned to baseline values at 24 °C (Figure 4D–F) and that heat not only induced an outward current but also increased the conductance through TREK channels in all three neuronal subtypes, as detailed in Table 4.

To determine the I-V relationships of heat-induced currents, voltage ramps from −30 to −100 mV (10 mV/s) were applied to A- (*n* = 5), Ah- (*n* = 6) and C-type (*n* = 5) neurons in the presence of cocktail A, both before and after increasing the temperature. The temperature-sensitive current was obtained by subtracting the control trace from a heat-activated trace. These heat-induced currents, presumably mediated by TREK channels, exhibited similar reversal potentials and no apparent voltage dependence across the three neuronal types (Figure 4G–I).

To ensure the involvement of TREK channels in the heat-induced current, the same experiments were repeated in the presence of cocktail B, which blocks the same channels as cocktail A plus several TRPs and Ca^+2^-activated K^+^ channels (see the Section 4). Upon increasing the temperature, an outward current and a fall in membrane resistance were clearly observed at −30 mV in all three neuronal types. The change in the conductance from room to physiological temperature was statistically significant in A-, Ah-, and C-type neurons (paired *t*-test, *p* < 0.001). Predictably, heat-induced currents were inhibited by 100 µM of fluoxetine in 10 A-type, 16 Ah-type, and 9 C-type neurons (Figure 5A–C). The current elicited by voltage ramps at 24 °C was subtracted from that at 37 °C to isolate the temperature-activated component. Voltage ramps were performed in three A-, five Ah-, and five C-type neurons (Figure 5D–F) to identify I-V relationships, and temperature-dependent currents exhibited similar reversal potentials in the three neuronal subtypes. The results were consistent with those obtained using cocktail A and are summarized in Table 4.

### 2.5. Open Probability of TREK Increases at Physiological Temperature

To study the effect of temperature at the single-channel level, we performed cell-attached recordings in NG neurons. First, voltage steps ranging from −100 to 100 mV at RT were applied to generate an I-V curve to further characterize and classify the recorded ion channels. From all the patched channels (109 recordings), 19 resulted in TREK channels. 11 (57.9% of total channels recorded) exhibited I-V curves similar to those of TREK1 channels, showing their typical outward rectification [24] (Figure 6A), while 3 channels (15.8%) exhibited I-V curves resembling those TREK2 channels, displaying the characteristic inward rectification [25] (Figure 6B); and 5 channels (26.3%) showed I-V curves similar to TRAAK channels with no rectification [26,27] (Figure 6C). The membrane potential was then set to −60 mV, close to the RMP of these cells [6,28]. This potential helps maintain the hyperpolarized state, minimizing interference from other voltage-dependent currents and enabling more precise single-channel measurements. Amplitudes were measured and the conductance of each channel was calculated using Ohm’s Law (I = V/R; G = 1/R; G = I/V). Channels previously classified as TREK1 showed a conductance of 111.70 ± 5.24 pS when the membrane potential was clamped at −60 mV and at RT. Channels classified as TREK2 exhibited a conductance of 128 ± 1.09 pS, while TRAAK channels showed a conductance of 117.7 ± 2.72 pS. Similar conductance values and current amplitudes for TREK1, TREK2, and TRAAK channels under the same conditions have been reported in the literature [29,30,31].

Subsequently, the temperature of the extracellular solution was raised to physiological levels, resulting in an increase in channel activity (Figure 6D–F). Furthermore, amplitude (Figure 6G), open probability (Po, Figure 6H), and open dwell time (Figure 6I) were measured one minute before and after the temperature increased. The temperature did not affect the current amplitude of TREK1 (paired *t*-test, *p* > 0.05, *n* = 11) nor TREK2 (paired *t*-test, *p* > 0.05, *n* = 3) channels, but a slightly increased TRAAK amplitude (paired *t*-test, *p* < 0.05; *n* = 5, Figure 6H). In agreement with previous studies, the TREK channels recorded exhibited large conductance values under symmetrical conditions, consistent with the biophysical properties of TREK channels [32,33] that facilitate efficient K^+^ efflux and rapid adaptation to mechanical or chemical stimuli, representing a hallmark activity. Although no changes in amplitude were observed, raising the temperature increased the Po in TREK1 (paired *t*-test, *p* < 0.001; *n* = 11), TREK2 (paired *t*-test, *p* < 0.05, *n* = 3), and TRAAK channels (paired *t*-test, *p* < 0.05; *n* = 5, Figure 6H) and the open dwell time was reduced in TREK1 channels (paired *t*-test, *p* < 0.001; *n* = 11), TREK2 channels (paired *t*-test, *p* < 0.05, *n* = 3), and TRAAK channels (paired *t*-test, *p* < 0.01, *n* = 5, Figure 6I). All populational data are detailed in Table 5. Taken together, these findings highlight a plausible increase in TREK subfamily activity at physiological temperature, suggesting a potential role for these channels in thermal homeostasis control.

## 3. Discussion

In the present work, we demonstrate that TREK channels present in mouse NG are sensitive to temperature changes, suggesting that TREK currents which participate in the modulation of internal body temperature can also modulate the behavior of nodose neurons.

TRPs have been generally considered the primary mediators of thermal signal transduction to the central nervous system (CNS). When a thermal noxious stimulus is perceived, the opening of TRPs provokes a depolarization in sensory neurons. In particular, a non-specific cation channel, mainly TRPV1, has been identified as a key heat-sensitive channel involved in heat-induced pain perception [1,32,33]. Recent studies have also demonstrated that the TREK subfamily of K2P channels is also temperature-sensitive and may serve a protective role by counteracting the overactivation of excitatory (depolarizing) channels [16,34,35,36]. In fact, in this study, we observed that TREK-like currents are responsible for partially counteracting the excitatory effects of depolarizing channels on the firing rate of NG neurons. Interestingly, TREK1 channels are co-expressed with TRPV1 in both small- and medium-diameter sensory neurons of the DRG and trigeminal ganglia [35,37,38,39], suggesting a counterbalancing mechanism in membrane potential regulation. However, despite the molecular similarities between DRG and NG neurons, functional differences may still be present. These differences are challenging to assess in vitro, as primary culture systems may not accurately reflect the complexity observed in vivo.

We managed to isolate TREK currents in NG neurons by using a mixture of blockers for Na^+^, Ca^2+^, classic K^+^, TRPV, TRPC5, and Ca^2+^-dependent K^+^ channels and for h-current. Under these conditions, increasing the temperature from 24 to 37 °C elicited an outward current accompanied by an increase in membrane conductance, both of which were blocked by fluoxetine. TREK currents also increase when the temperature rises from 24 °C to 37–42 °C in DRG neurons, where TREK1 channels are highly expressed [26,35]. Otherwise, the temperature sensitivity of TREK1 channels expressed in oocytes and COS cells is higher than that of other K2P channels [16]. In fact, the current through TREK1 channels expressed in *Xenopus* oocytes is absent at 12° C and becomes a large outwardly rectifying current at 37 °C [27]. Indeed, the maximum temperature sensitivity of TREK1 is observed between 32 °C and 37 °C, with a 9-fold increase in current amplitude per 10 °C [27]. Furthermore, TREK1 exhibits its peak average activation at 37 °C, which corresponds to the physiological temperature of most mammals [27]. It is worth noting that TREK channels remain responsive to other stimuli—such as arachidonic acid, pH changes, and membrane stretch—even when activated by temperature, as demonstrated in single-channel recordings at 37° C [27].

Neurons from mouse NG predominantly express TRESK and TREK1 K2P channels [13]. TRESK channels are opened at RT but are not severely affected by changes in temperature [15,25,40,41,42]. In contrast, TREK1 channel, the second most highly expressed K2P channel in mouse NG neurons [13], is strongly associated with temperature sensitivity. Specifically, the TREK1 current increases substantially when temperature rises from RT to physiological values [37]. Similar behavior is observed in the other TREK subfamily subunits, TREK2 and TRAAK channels [1,15,16,43]. It has been shown that TREK2 regulates thermic perception in moderate temperature ranges in DRG neurons [44], and that its activity increases when temperature increases in heterologous systems [15,45]. TRAAK channels [15] are involved in hyperalgesia, presenting a higher activation threshold when temperature increases [15], but they do not respond to cold perception [46]. A recent study showed that this subfamily does not present a significant activation with temperature changes compared to TREK1 and TREK2 [47]. It seems that the three members of the TREK subfamily play a complementary role in thermosensation. Despite this, expression levels lead us to think that TREK1 is the main contributor in controlling neuronal excitability when temperature is modified in mouse NG neurons. Our findings show that a current through TREK channels is activated even at temperatures that are close to physiological temperatures in mouse NG neurons. In fact, the hyperpolarizing effect of TREK channels on the RMP is well documented, suggesting a neuroprotective function [14].

Interestingly, TREK1 knock-out (KO) mice, as well as mice lacking both TREK1 and TRAAK, exhibit heat hyperalgesia. On the contrary, only the double-KO, but not the single-KO, mice display cold hyperalgesia [16]. Our findings support the idea of numerous investigations highlighting the significant function of TREK channels in thermal perception and their protective role. In this study, we reported the modulation of TREK channels by heat due to changes in the open probability of single TREK channels at temperatures close to physiological. In conclusion, the increase in temperature from 24 °C to 37 °C strongly activates a TREK 1-like current in all three types of NG neurons, suggesting that TREK channels are likely implicated in thermal transduction from the core to the CNS, potentially contributing to the mechanisms that maintain homeostasis.

## 4. Materials and Methods

All experimental procedures were performed using a total of 250 male and female 30- to 90-day-old CD-1 mice, housed in the Bioexperimentation Centre of the University of Vigo. Animals had ad libitum access to commercial food and water and were maintained under controlled environmental conditions (temperature: 20–24 °C; relative humidity: 45–55%) with a 12 h light/dark cycle. All animal experiments were authorized by the Spanish (RD 53/2013) and European (2010/63/EU) Research Councils, as well as supervised by the Scientific Committee of the University of Vigo, following the European and Spanish animal protection directives.

### 4.1. Cell Culture

NG neurons were obtained as previously described [48,49]. Briefly, mice were deeply anesthetized with CO_2_ and immediately decapitated. NG were removed, cleaned, and incubated in collagenase (3 mg/mL) and trypsin (1 mg/mL) enzymatic solutions for 15 and 30 min, respectively. Subsequently, mechanical disaggregation was carried out using Pasteur pipettes and cells were centrifuged (1600 rpm, 20 °C) for 3 min, supernatant was discarded, and pellet was resuspended and plated into 35 mm dishes previously treated with laminin (10 µg/mL for 2 h). Cells were cultured in fresh medium containing a L-15 solution supplemented with 10% fetal bovine serum, 24 mM NaHCO_3_, 38 mM D-glucose, 100 UI/mL penicillin–100 μg/mL streptomycin, 2 mM L-glutamine and 50 ng/mL nerve growth factor, which were added to each dish. Finally, cells were maintained in an incubator with 5% CO_2_ at 37 °C for 24 h.

### 4.2. Electrophysiology

Whole-cell patch-clamp recordings (perforated patch configuration) were performed 24 h after primary neuronal cultures were established. Artificial cerebrospinal fluid (ACSF) was continuously perfused at approximately 10 mL/min and contained (in mM): NaCl 140, KCl 3, MgCl_2_ 1, CaCl_2_ 2, D-glucose 10, HEPES 10, gassed with synthetic air and a pH of 7.2 adjusted with Tris (Tris (hydroxymethyl)-amino-methane) (osmolarity 270–300 mOsm/L). Borosilicate glass patch pipettes were obtained with a range tip resistance of 4–6 MΩ, using a P-1000 micropipette puller (Sutter Instruments, Novato, CA, USA). Patches with series resistance of >25 MΩ were discarded, and junction potentials of less than 10 mV were not corrected. The intracellular solution contained 90 mM K-acetate, 20 mM KCl, 3 mM MgCl_2_, 1 mM CaCl_2_, 3 mM EGTA, and 40 mM HEPES NaOH 1 to adjust pH to 7.2 (osmolality 270–300 mOsm/L). Amphotericin-B (75 µg/mL) was added to the intracellular solution for the perforated whole-cell variant.

To perform in vitro electrophysiological experiments, an Axopatch 200B (Molecular Devices, San Jose, CA, USA) amplifier was used along with a Digidata 1440A and the PClamp 10.0 software (both from Molecular Devices, San Jose, CA, USA). Signals were acquired at 10 KHz and filtered at 5 kHz for current-clamp experiments and acquired at 5 kHz and filtered at 1 kHz for voltage-clamp experiments.

For NG neurons classification, in the voltage-clamp configuration, neurons were clamped at −30 mV and capsaicin 1 µM was added to the bath solution. Afterwards, 500 ms voltage steps from −60 to 0 mV were applied in the presence of CdCl_2_ 100 µM to block cationic currents before and after TTX 0.5 µM was added to block Na^+^ voltage-dependent channels and cationic currents, respectively. Three cell types with different electrophysiological and pharmacological profiles are present in NG neurons [6]: A-type neurons which do not respond to CAP and are without TTX-resistant Na^+^ currents (CAP−; TTX-S); Ah-type neurons, which do not respond to CAP but exhibit TTX-resistant Na^+^ currents (CAP−; TTX-SR); and finally C-type neurons, which show an inward current in response to capsaicin and TTX-resistant Na^+^ currents (CAP+; TTX-SR). This classification was performed before temperature experiments for each cell, and drugs were thoroughly washed out before proceeding with other protocols.

For temperature experiments, a heater system (Warner Instrument, Holliston, MA, USA) was used to increase the bath solution temperature while monitoring it. The bath solution perfusion system temperature was increased and a thermosensor was placed in the cultured well, providing real-time feedback to allow temperature deviations to be corrected. In current-clamp mode and after fixing the neurons at −60 mV, temperature ramps from 24 to 50 °C were applied to observe changes in membrane potential in a gap-free configuration. Next, 100 µM of fluoxetine and 1 µM spadin, two TREK channel blockers, were added to the bath solution to observe changes in membrane potential in response to temperature variations. Fluoxetine is an antidepressant that effectively blocks TREK channels [50,51,52] and spadin selectively inhibits TREK1 channel current but fails to inhibit TREK2 and TRAAK subtypes [20,21]. In another batch of experiments, APs were discharged by injecting increment current pulses from −50 to 350 pA (50 pA increments, 1 s each) before and after raising the temperature to 37 °C to study differences in the firing rate. The experiment was replicated by adding 1 µM spadin to inhibit the TREK channel involvement. Furthermore, AP individual parameters were measured at 24 and 37 °C as follows: latency was measured from the beginning of the stimulus to the AP-positive peak; amplitude was determined from the most positive to the most negative voltage of the AP; the duration was the 50% amplitude width; AHP was calculated from the resting potential to the most negative voltage; and finally, the frequency was calculated between the first and the second AP fired.

In the voltage-clamp configuration and after fixing the voltage at −30 mV, a cocktail of ion channels blockers (cocktail A) was added to the bath solution to isolate TREK currents and temperature ramps were performed. Cocktail A contained TTX 0.5 μM, CdCl_2_ 100 μM, CsCl 1 mM, TEA 15 mM, and 4-aminopyridine (4-AP) 2 mM to block voltage-dependent sodium, potassium, cationic, and calcium currents. To ensure the isolation of TREK currents, the same experiments were carried out in the presence of cocktail B, which contained the same channel blockers as cocktail A plus capsazepine 1 µM (a TRPV1 channel blocker), apamin 200 nM (small conductance Ca^+2^ activated K^+^ channel blocker), paxillin 1 µM (large conductance Ca^+2^ activated K^+^ channel blocker), clemizole 10 µM (TRPC5 channel blocker), and ruthenium red 10 µM (TRPs blocker). Other voltage-clamp recordings (V_clamp_ = −30 mV) were performed by applying 15 mV negative pulses every 50 ms to calculate membrane conductance. Membrane conductance (G) was estimated based on Ohm’s Law, where G = 1/R, so G = I/V, where R is the resistance, I is the current, and V is the voltage. Furthermore, voltage ramps from −30 to −100 mV (10 mV/s) were also applied at 24 and 37 °C to subtract the temperature-sensitive voltage–current (I-V) relationship.

For single-channel analysis, cell-attached recordings were obtained, and identical bath and pipette solutions were used, containing (mM) 150 KCl, 1 MgCl_2_, 5 EGTA, and 10 HEPES. pH 7.2 was achieved with KOH 1 mM and osmolarity between 290 and 310 mOsm. Wax-coated pipettes achieved resistances above 10 MΩ. Voltage steps from −100 to 100 mV were applied and I-V curves constructed to characterize TREK subfamily members. Neurons were clamped at −60 mV and temperature ramps were applied to measure the amplitudes of each channel before and after the temperature increase. In another set of experiments, a voltage-clamp step protocol from −100 to 100 mV was applied before and after raising the temperature. Data were sampled at 20 kHz and low-pass filtered at 2 kHz using the amplifier’s built-in filter. Open probability (Po), open dwell time, and amplitude were measured to determine differences between RT and physiological temperature in each TREK channel subtype. Channel openings faster than 50 µs were discarded. The Po was calculated using the equation Po = to/T, where Po is the open-state probability of channels (N is the number of the channels), to is the total time in which the channel was found in the open state, and T is the total observation time.

pClamp 10.0 and Origin 9.0 software were used to measure, analyze, and plot data. Averages were represented as mean ± SEM and statistical differences were assessed using a paired-sample *t*-test or one-way and two-way ANOVA, and Tukey’s test for multiple comparisons among three neuronal groups. Differences were considered significant when *p* < 0.05 (* or #), *p* < 0.01 (* or #) or *p* < 0.001 (* or #).

## Figures and Tables

**Figure 1 ijms-26-09119-f001:**
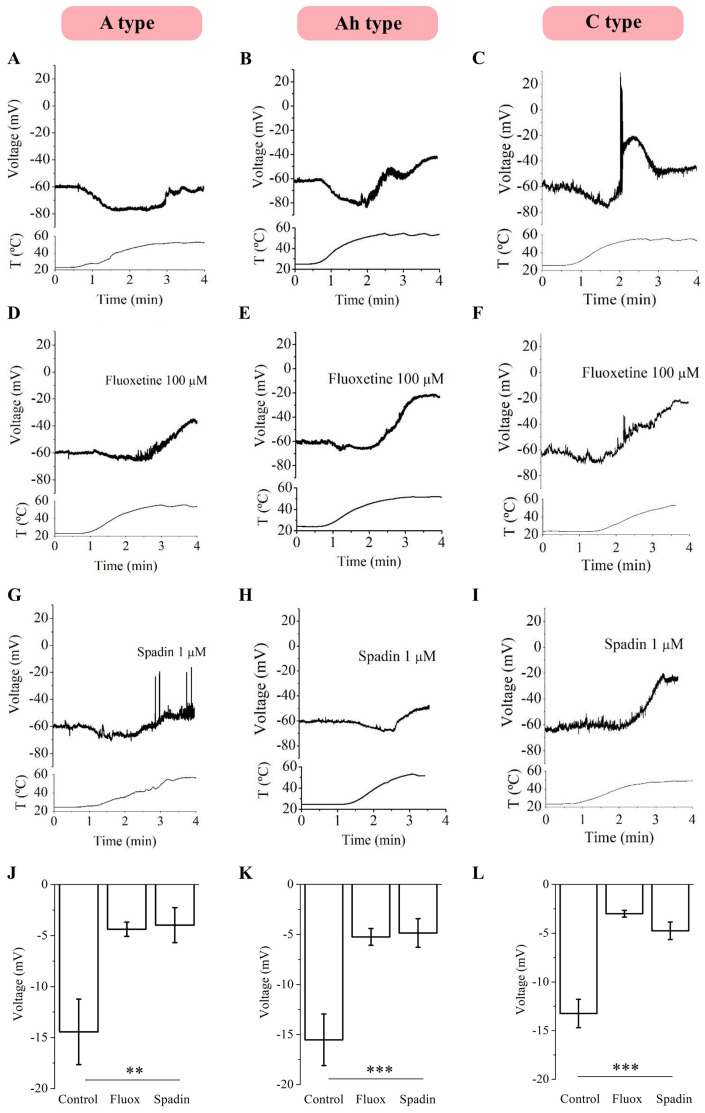
Effects of increasing temperature on the RMP of mNG neurons in the presence and in the absence of fluoxetine and spadin. Increasing temperatures from 24 °C to 50 °C induced a hyperpolarization of the membrane potential in A- (*n* = 7), Ah- (*n* = 10) and C-type (*n* = 8) neurons (**A**–**C**) at 37 °C. Heat-induced hyperpolarization is inhibited by 100 µM of fluoxetine (**D**–**F**) and 1 µM spadin (**G**–**I**) in A, Ah, and C neurons ((**J**–**L**), One-way ANOVA, ** *p* < 0.01; *** *p* < 0.001). T = Temperature.

**Figure 2 ijms-26-09119-f002:**
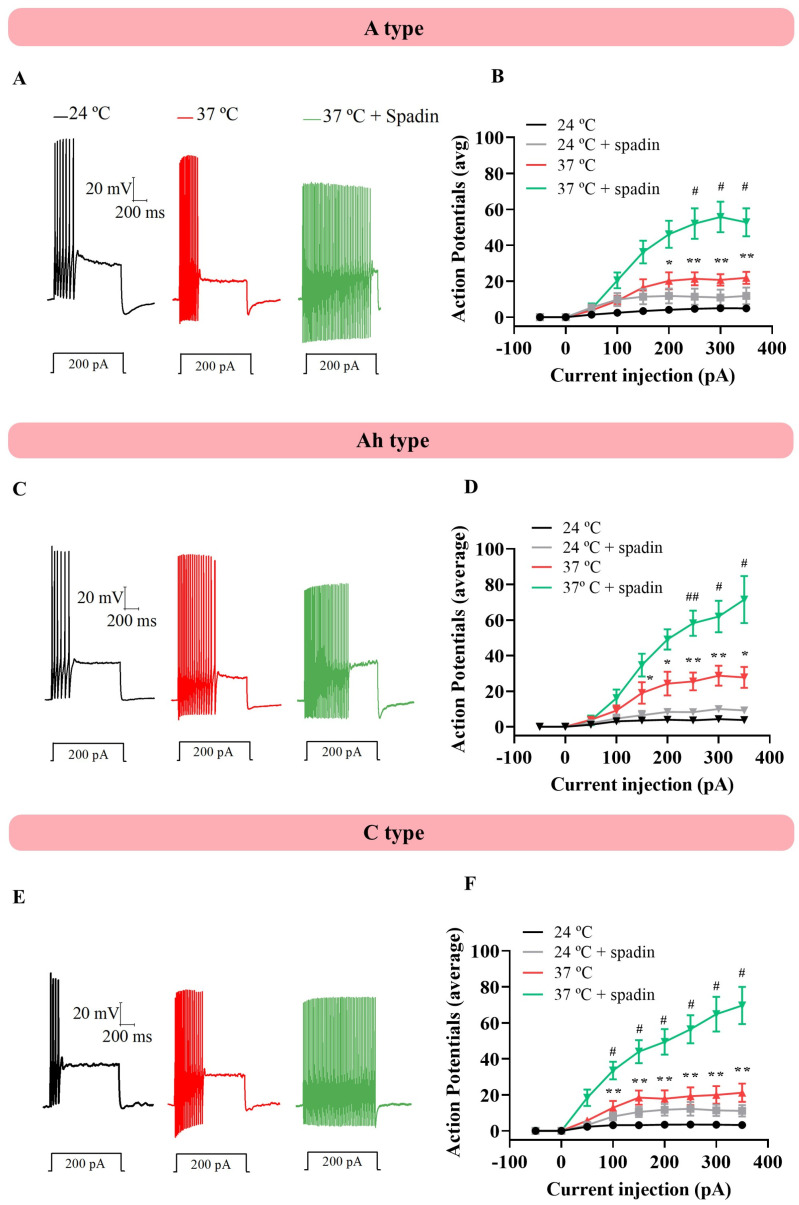
Physiological temperature generated changes in cell excitability. (**A**) Current increment injections from −50 to 350 (I50, 1 s each) were applied when the temperature was increased from 24 °C to 37 °C in the absence and presence of 1 µM spadin. (**B**) A type neurons (*n* = 10)) presented a higher firing rate at physiological values, which significantly increased when the TREK1 channel was blocked by 1 µM spadin (two-way ANOVA with multiple comparisons). (**C**,**D**) After applying the same protocol in Ah neurons and (**E**,**F**) C type neurons, the increase in number of action potencials was observed, with no differences between neuronal types. An asterisk (*) indicates a statistically significant difference in AP number between 24 °C and 37 °C. A hashtag (#) indicates a statistically significant difference in the number of APs at 37 °C compared to 37 °C in the presence of 1 µM spadin. # *p* < 0.05, ## *p* < 0.01, * *p* < 0.05 and ** *p* < 0.01.

**Figure 3 ijms-26-09119-f003:**
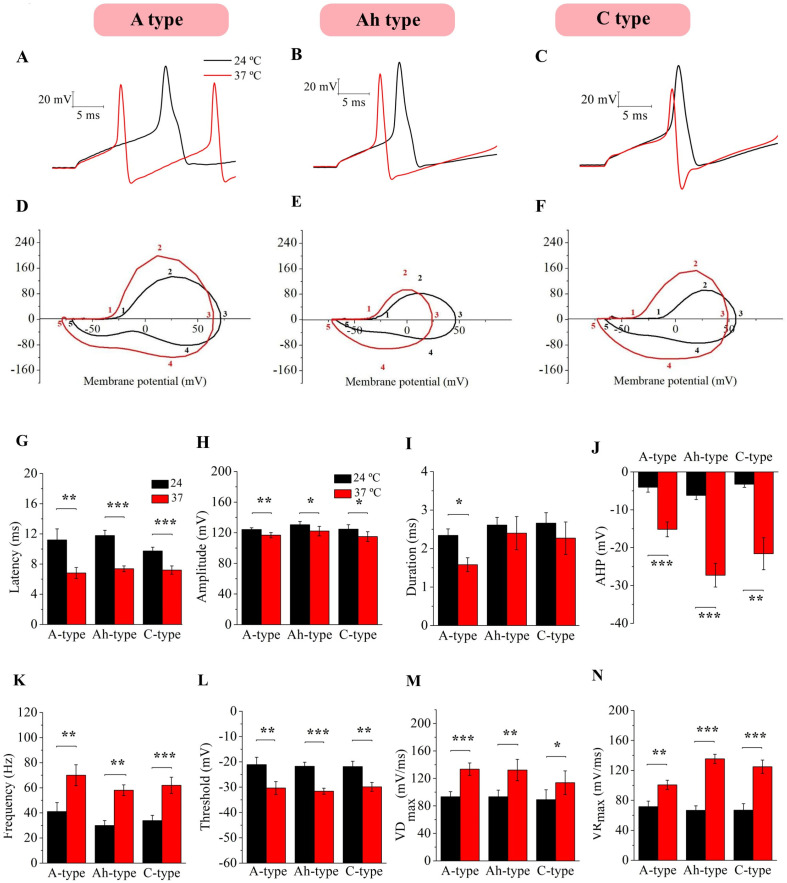
Changes in individual action potential parameters in response to temperature. Individual AP parameters were measured at 24 and 37 °C for each subtype (**A**–**C**) and AP phase plots were constructed for A (**D**), Ah (**E**), and C neurons (**F**) at 24 °C (black trace) and 37 °C (red trace) to study changes in velocity. As observed in the representative traces of the three neuronal types, latency (**G**) and amplitude (**H**) significantly decreased, duration (**I**) changed only in A-type neurons, and AHP (**J**) and frequency (**K**) were significantly augmented in A (*n* = 10), Ah (*n* = 10), and C neurons (*n* = 10) when the temperature was increased at 37 °C (paired, *t*-test, * *p* < 0.05, ** *p* < 0.01 and *** *p* < 0.001). The threshold (**L**), maximal velocity of depolarization (VDmax, (**M**)), and maximal velocity of repolarization (VRmax, (**N**)) showed different results at 37 °C compared to room temperature in all three neural types (paired, *t*-test, * *p* < 0.05, ** *p* < 0.01 and *** *p* < 0.001).

**Figure 4 ijms-26-09119-f004:**
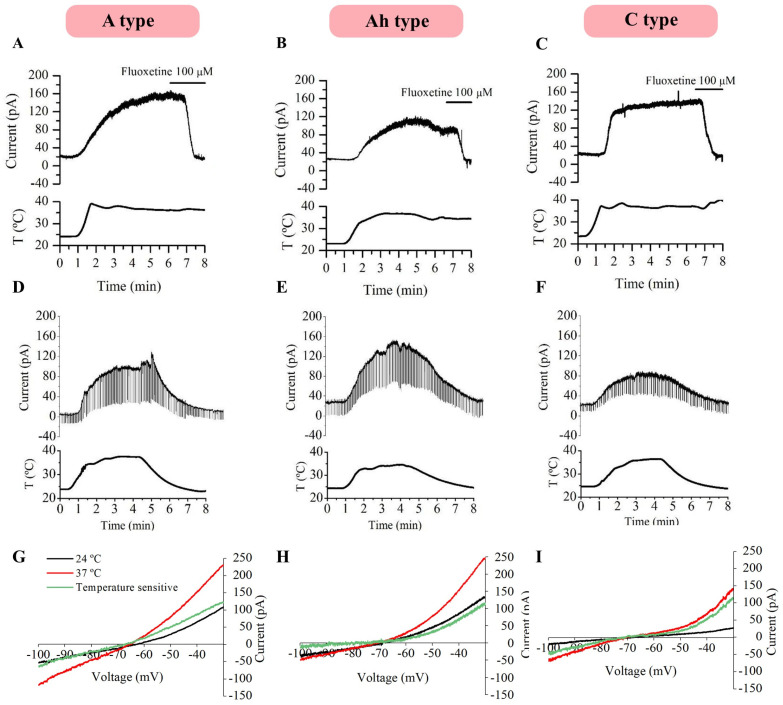
Heat-activated outward current in the presence of cocktail A. Heat-induced (from 24 to 37° C) outward currents in the presence of cocktail A were strongly blocked with 100 μM of fluoxetine in all three neural types: A-type (*n* = 5, (**A**)), Ah-type (*n* = 9, (**B**)) and C-type neurons (*n* = 10, (**C**)). The outward currents activated by increasing the temperature were reversibly abolished upon the return to room temperature (24 °C) in all cell types. In this case, voltage steps (−50 mV every 50 ms) were applied at 50 Hz and an increase in membrane conductance was observed (**D**–**F**). I-V relationships for heat-induced currents in A- (**G**), Ah- (**H**) and C-neurons (**I**) obtained by subtracting the currents at 24 °C (black trace) and 37 °C (red trace) after the application of a voltage ramp. T: Temperature.

**Figure 5 ijms-26-09119-f005:**
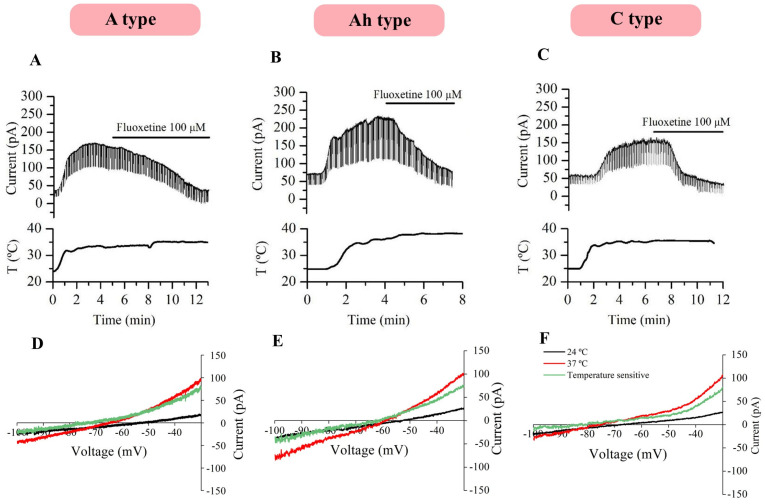
Heat-activated outward current in the presence of cocktail B. Heat-induced currents inhibited by 100 µM of fluoxetine in A- (**A**), Ah- (**B**) and C-type neurons (**C**) in the presence of cocktail B. An increase in conductance at 37 °C was observed in three neuronal subtypes. Temperature-activated currents (green trace) were obtained by subtracting the elicited current at 24 °C from the current activated at 37 °C (**D**–**F**). T: Temperature.

**Figure 6 ijms-26-09119-f006:**
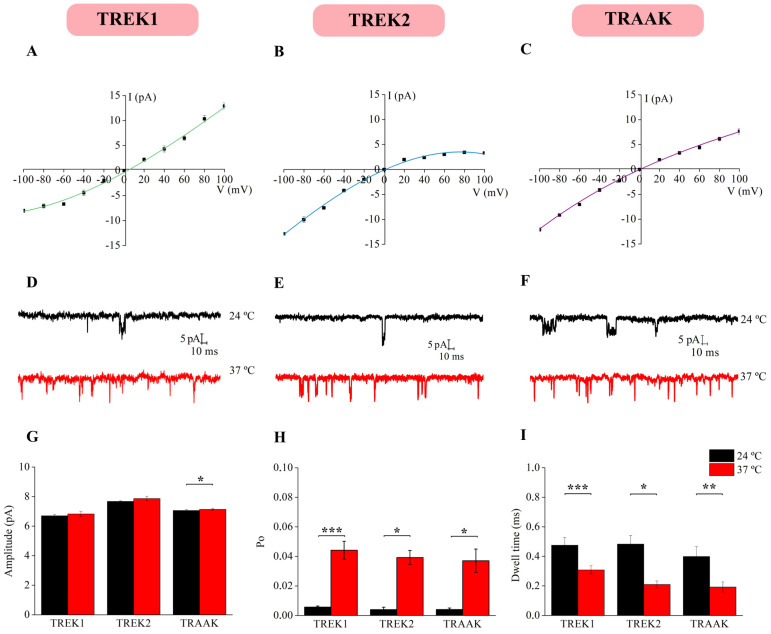
TREK subfamily single-channel properties. Under symmetrical K^+^ concentrations and using cell-attached configuration, current–voltage relationships were constructed, and a total of 11 TREK-1 (**A**), 3 TREK2 (**B**) and 5 TRAAK channels (**C**) were identified. Single-channel recordings at −60 mV revealed increases in activity at 37 °C for TREK-1 (**D**), TREK-2 (**E**), and TRAAK (**F**). The single-channel current amplitude (**G**), Po (**H**), and dwell time (**I**) were significantly different at room (24 °C) and physiological (37 °C) temperatures (paired *t*-test, * *p* < 0.01, ** *p* < 0.01 and *** *p* < 0.001).

**Table 1 ijms-26-09119-t001:** Heat-induced hyperpolarization (37 °C) and depolarization (50 °C) in the three NG neuronal types in the absence and presence of 100 µM fluoxetine and 1 µM spadin.

	A-Type (mV)		Ah-Type (mV)		C-Type (mV)	
	37 °C	50 °C	37 °C	50 °C	37 °C	50 °C
**Control**	−10.8 ± 1.1 (*n* = 7)	30.4 ± 5.5 (*n* = 7)	−9.3 ± 1 (*n* = 10)	29.36 ± 8.9 (*n* = 10)	−8.2 ± 1 (*n* = 8)	28.53 ± 3.1 (*n* = 8)
**Fluoxetine**	−2.74 ± 0.4 (*n* = 5)	20.7 ± 1.2 (*n* = 5)	−4.7 ± 0.7 (*n* = 12)	20.71 ± 1.3 (*n* = 12)	−2.2 ± 0.36 (*n* = 7)	17.94 ± 1.8 (*n* = 7)
**Spadin**	−2.17 ± 0.31 (*n* = 8)	17.67 ± 6.7 (*n* = 8)	−2.89 ± 0.4 (*n* = 6)	17.27 ± 2.9 (*n* = 6)	2.03 ± 0.2 (*n* = 4)	21.2 ± 3.8 (*n* = 4)

**Table 2 ijms-26-09119-t002:** Number of action potentials in the three different types of NG neurons before and after a temperature increase in the absence and presence of 1 µM of spadin (Sp).

Current Injected (pA)	A-Type				Ah-Type				C-Type			
	24 °C	24 °C + Sp	37 °C	37 °C + Sp	24 °C	24 °C + Sp	37 °C	37 °C + Sp	24 °C	24 °C + Sp	37 °C	37 °C + Sp
	*n* = 10	*n* = 7	*n* = 10	*n* = 7	*n* = 10	*n* = 10	*n* = 10	*n* = 10	*n* = 10	*n* = 7	*n* = 10	*n* = 7
**50**	1.4 ± 0.58	3.7 ± 1.5	3.8 ± 1.12	2.1 ± 1.9	1.2 ± 0.3	2 ± 1.2	4 ± 1.3	3.6 ± 2.7	2.3 ± 0.5	3.3 ± 1.1	5.7 ± 1.67	18.4 ± 4.5
**100**	2.5 ± 0.73	7.5 ± 3.6	9.1 ± 2.83	18.1 ± 4.2	3.1 ± 0.8	4.6 ± 2.3	9.1 ± 2.9	16.1 ± 4.8	3.2 ± 0.74	8 ± 2.2	12.9 ± 3.79	33.6 ± 4.9
**150**	3.4 ± 0.89	8.8 ± 4.2	16.6 ± 4.54	34.6 ± 7.1	3.5 ± 0.7	6.5 ± 2.5	19 ± 6.1	34.7 ± 6.4	3.2 ± 0.63	10.6 ± 2.9	18.5 ± 3.98	44 ± 3.4
**200**	4.1 ± 1.16	9.5 ± 4.5	20.2 ± 4.74	44.9 ± 8.6	1.4 ± 0.7	8.4 ± 1.8	24.3 ± 6.7	49.1 ± 5.7	3.4 ± 0.56	11.7 ± 3.3	17.9 ± 4.62	49.4 ± 7.1
**250**	4.7 ± 1.47	9.5 ± 5.2	21.4 ± 3.63	50.9 ± 9.7	3.6 ± 0.7	8.3 ± 1.4	25.5 ± 5	58.2 ± 7.1	3.5 ± 0.62	12.3 ± 3.7	19.3 ± 4.88	56.4 ± 7.8
**300**	5 ± 1.50	9.3 ± 5.3	20.8 ± 3.2	55 ± 9.8	4.4 ± 0.9	10 ± 1.9	28.7 ± 5.6	62 ± 8.8	3.4 ± 0.06	11.4 ± 3.2	20 ± 4.9	64.9 ± 9.6
**350**	4.9 ± 1.55	9.8 ± 5.4	21.9 ± 3.3	53 ± 9	3.8 ± 0.6	9.2 ± 1.9	27.8 ± 5.9	71.5 ± 13.2	3.3 ± 0.54	11.4 ± 3.2	21.2 ± 5.05	69.7 ± 10.3

**Table 3 ijms-26-09119-t003:** Electrophysiological parameters of the three types of NG neurons before and after raising the temperature.

Individual Parameters	A-Type (*n* = 10)		Ah-Type (*n* = 10)		C-Type (*n* = 10)	
	24 °C	37 °C	24 °C	37 °C	24 °C	37 °C
**Amplitude (mV)**	124.24 ± 2.2	116.83 ± 3.32 **	130.4 ± 4.29	122.22 ± 6.23 *	124.69 ± 5.74	115.01 ± 6.29 *
**AP halfwidth (ms)**	2.34 ± 0.17	1.58 ± 0.10 *	2.61 ± 0.20	2.40 ± 0.43	2.66 ± 0.27	2.27 ± 0.42
**AHP (mV)**	−4.05 ± 1.28	−15.17 ± 1.95 ***	−6.19 ± 1.11	−27.29 ± 3.12 ***	−3.25 ± 0.81	−21.60 ± 4.24 **
**Threshold (mV)**	−21.09 ± 2.90	−30.35 ± 2.57 **	−21.76 ± 1.62	−31.60 ± 1.18 ***	−21.−85 ± 2.10	−29.88 ± 1.76 **
**VDmax (mV/ms)**	95.35 ± 7.37	133.47 ± 9.12 ***	93.40 ± 9.61	132.22 ± 15.62 **	89.07 ± 14.49	113.77 ± 17.13 *
**VRmax (mV/ms)**	−71.56 ± 7.46	−100.70 ± 6.12 **	−66.66 ± 5.95	−135.38 ± 6.13 ***	−66.95 ± 8.65	−124.90 ± 8.90 ***

* *p* < 0.05, ** *p* < 0.01, *** *p* < 0.001.

**Table 4 ijms-26-09119-t004:** Heat-induced current and conductance at 24 and 37 °C in the three NG neuronal types.

	A-Type	Ah-Type	C-Type
**COCKTAIL A**			
**Heat-induced current (pA)**	82.20 ± 5.49 (*n* = 18)	111.75 ± 9.52 (*n* = 26)	72.66 ± 7.60 (*n* = 21)
**Conductance at 24 °C (nS)**	2.13 ± 0.24 (*n* = 5)	2.02 ± 0.18 (*n* = 13)	1.98 ± 0.12 (*n* = 10)
**Conductance at 37 °C (nS)**	3.25 ± 0.29 (*n* = 5) ***	5.59 ± 0.66 (*n* = 13) ***	3.30 ± 0.24 (*n* = 10) ***
**COCKTAIL B**			
**Heat-induced current (pA)**	65.91 ± 8.41 (*n* = 13)	110.51 ± 10.46 (*n* = 21)	72.60 ± 10.80 (*n* = 14)
**Conductance at 24 °C (nS)**	2.09 ± 0.20 (*n* = 10) ***	2.30 ± 0.19 (*n* = 16) ***	2.18 ± 0.17 (*n* = 9) ***
**Conductance at 37 °C (nS)**	3.82 ± 0.37 (*n* = 10) ***	5.24 ± 0.39 (*n* = 16) ***	3.73 ± 0.39 (*n* = 9) ***

*** *p* < 0.001.

**Table 5 ijms-26-09119-t005:** Single-channel characteristics of the three members of K2P channels at 24 and 37 ◦C.

	TREK1 (*n* = 11)		TREK2 (*n* = 3)		TRAAK (*n* = 5)	
	24 °C	37 °C	24 °C	37 °C	24 °C	37 °C
**Po**	0.006 ± 0.002	0.044 ± 0.02 ***	0.004 ± 0.002	0.04 ± 0.008 *	0.004 ± 0.002	0.037 ± 0.017 *
**Dwell time (ms)**	0.475 ± 0.17	0.308 ± 0.092 ***	0.483 ± 0.1	0.21 ± 0.04 *	0.4 ± 0.15	0.19 ± 0.07 **
**Amplitude (pA)**	6.70 ± 0.31	6.82 ± 0.56	7.68 ± 0.065	7.87 ± 0.27	7.06 ± 0.16	7.14 ± 0.15 *
**Conductance (pS)**	111.7 ± 5.2	113.7 ± 9.3	128 ± 1.1	131.2 ± 4.5	117.7 ± 2.7	119 ± 2.6

* *p* < 0.05, ** *p* < 0.01 and *** *p* < 0.001.

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
