# Peer review of "Increasing Temperature Activates TREK Potassium Currents in Vagal Afferent Neurons from the Nodose Ganglion"

_ijms, 2025, doi:10.3390/ijms26189119_

Round 1

Reviewer 1 Report

Comments and Suggestions for Authors

Existing research has established the involvement of TRP channels in thermal transduction; however, it is also recognized that TREK channels contribute to signal transduction. This study effectively organizes findings related to each subtype in NG neurons and employs various experimental approaches, including whole-cell and single-channel recordings.

I would like to propose a few additional points for consideration:

  1. While it is widely accepted that TRP channels play a role in thermal transduction, I am interested in whether this aspect has been entirely excluded in the measurement of TREK channels. It would be valuable to explore whether similar results are obtained when a TRP-specific inhibitor is applied under the same experimental conditions.
  2. The study illustrates the inhibitory effect of a specific TREK channel inhibitor. To further substantiate the findings, it would be beneficial to demonstrate that this inhibition is completely abolished when knockdown or knockout (KO) is achieved using CRISPR-Cas9 techniques.
  3. In Figure 6, the changes in single-channel properties are presented not only for TREK-1 but also for TREK-2 and TRAAK, another subfamily of TREK channels. While your data and previous studies indicate that TREK-1 is predominantly expressed in NG neurons, Figure 1 suggests that NG neurons still exhibit a slight decrease in membrane potential following TREK-1 blockade. This raises the question of whether TREK-2 and TRAAK might also be involved. It would be more accurate to describe the contribution of TREK-1 as "mainly responsible" rather than "specific to TREK-1." Additionally, providing data that demonstrates the lack of involvement of other TREK subfamilies in NG neuronal activities at elevated temperatures would clarify the implications of increased open probability for TREK-2 and TRAAK at physiological temperatures.
  4. Figure 6 presents the characteristics of each subtype; however, the sample size (n) for TREK-2 appears to be only three. Increasing the sample size for this data point would enhance the robustness of the findings.

Minor Comments

Lines 110-112: The content appears to overlap with the phrase on lines 104-106. It would improve clarity to remove this redundancy.

Line 226: The details of cocktail B are located in Table 4, so it is necessary to correct the reference from Table 5 to Table 4.

Figure 6 (Labels A, B, and C): The data in Figure 6 illustrate the single-channel properties of the TREK subfamily. However, the labels A, B, and C in the pink box currently describe NG neuronal subtypes. Please revise these labels to accurately reflect TREK-1, TREK-2, and TRAAK.

Author Response

Comment 1: While it is widely accepted that TRP channels play a role in thermal transduction, I am interested in whether this aspect has been entirely excluded in the measurement of TREK channels. It would be valuable to explore whether similar results are obtained when a TRP-specific inhibitor is applied under the same experimental conditions.

Response 1: Thank you for your valuable comment. Indeed, we decided to entirely focus this work in studying the contribution of TREK channels in thermal transduction rather than TRP channels. In section 2.2 we observed that, when increasing temperature from 24 to 50 ºC, a hyperpolarization followed by a depolarization of the membrane potential occurred under current clamp configuration. It has been established that TREK and TRP channels play opposite roles when temperature is increased. While TREK channels try to control the excitability by hyperpolarizing the membrane potential, TRP channels depolarize the membrane potential in order to initiate the stimulus transduction (doi:  10.3390/ijms20102371). For this reason, we decided to focus on the heat-induced hyperpolarizaton, and we wanted to demonstrate that this was caused by TREK channels by blocking them. Furthermore, this blockage was exclusively targeting the hyperpolarization (Figure 1, D-I). Also, when studying the heat induced outward current in the presence of cocktail B, different TRP channel blockers were used to further isolate TREK channel heat-activated current and demosrtate that it was indeed induced trough TREK channels. This has been clarified in Matherials and Methods section (lines 408 – 414).

Comment 2: The study illustrates the inhibitory effect of a specific TREK channel inhibitor. To further substantiate the findings, it would be beneficial to demonstrate that this inhibition is completely abolished when knockdown or knockout (KO) is achieved using CRISPR-Cas9 techniques.

Response 2: We thank the reviewer for the appreciation and agree it is an interesting point. In fact, it has been recently demonstrated that spadin reduces the astrocytic TREK1 currents and that, when knocking out TREK1 channels with a CRISPR/Cas9 approach, the spadin sensitive current are abolished, which means that spadin lost its effect when TREK1 is eliminated (doi: 10.3390/ijms21249639). Furthermore, a recent study showed that the protective effect of fluoxetine against decompression sickness was strongly reduced in TREK1 knockout mice, and that combined treatment with fluoxetine and spadin further mimicked the knockout phenotype, providing in vivo evidence that the effects of these inhibitors are indeed mediated through TREK1 channels (doi: 10.3389/fphys.2016.00042).

Comment 3: In Figure 6, the changes in single-channel properties are presented not only for TREK-1 but also for TREK-2 and TRAAK, another subfamily of TREK channels. While your data and previous studies indicate that TREK-1 is predominantly expressed in NG neurons, Figure 1 suggests that NG neurons still exhibit a slight decrease in membrane potential following TREK-1 blockade. This raises the question of whether TREK-2 and TRAAK might also be involved. It would be more accurate to describe the contribution of TREK-1 as "mainly responsible" rather than "specific to TREK-1." Additionally, providing data that demonstrates the lack of involvement of other TREK subfamilies in NG neuronal activities at elevated temperatures would clarify the implications of increased open probability for TREK-2 and TRAAK at physiological temperatures.

Response 3: Thank you for your valuable suggestions. Despite it has been proven that the three TREK subfamily members are involved in thermal sensation, previous results show that TREK1 is the mainly contributor to thermal transduction in mouse NG neurons. The discussion section has been modified (line 313-330) regarding this matter to properly discuss this problematic, as well as we have modified the text to talk about TREK1 as “mainly responsible”, as this reviewer has suggested.

Comment 4: Figure 6 presents the characteristics of each subtype; however, the sample size (n) for TREK-2 appears to be only three. Increasing the sample size for this data point would enhance the robustness of the findings.

Response 4: Thank you for this appreciation. We acknowledge that the sample size in our single-channel experiments is not sufficient to draw definitive conclusions. Nevertheless, our results indicate a predominance of TREK1, as it was the most frequently observed channel among the 109 patches analyzed, and this goes in line with our hypothesis and also with our previous findings (doi:  10.1007/s12031-012-9780-y). In primary cultures, identifying and recording from the channels of interest is inherently more challenging than in heterologous systems, where only the channels under study are expressed. This limitation explains the relatively small sample size in our work. Future studies specifically focused on single-channel properties would be valuable to strengthen and expand the current knowledge in this field.

Minor Comments

Comment 1: Lines 110-112: The content appears to overlap with the phrase on lines 104-106. It would improve clarity to remove this redundancy.

Response 1: Thank you. The sentence has been removed.

Comment 2: Line 226: The details of cocktail B are located in Table 4, so it is necessary to correct the reference from Table 5 to Table 4.

Response 2: Thank you. This mistake has been corrected.

Comment 3: Figure 6 (Labels A, B, and C): The data in Figure 6 illustrate the single-channel properties of the TREK subfamily. However, the labels A, B, and C in the pink box currently describe NG neuronal subtypes. Please revise these labels to accurately reflect TREK-1, TREK-2, and TRAAK.

Response 3: 

Thank you for the appreciation. Indeed, the labels were mistakenly placed there. A, B and C refers to TREK1, TREK2 and TRAAK. Pink boxes have been modified.

We would like to thank again the reviewer for the valuable comments. A new revised version of the mansucript, as well as modified figures has been uploaded.

Reviewer 2 Report

Comments and Suggestions for Authors

Temperature sensing is important in physiology to control body function. The manuscript “Increasing temperature activates TREK potassium currents in vagal afferent neurons from the nodose ganglion” submitted by Rueda-Ruzafa et. al. classified three nodose ganglia profiles in response to temperature shifts, voltage protocols as well as modulation by specific compounds. It is important and interesting to look next to the well described TRP channels at other channels. However, I have some problems to follow the story. Often important aspects are not or to late introduced. For example line 42 – “All three NG neuronal types…,” The three types are introduced in line 44. Another example is the used cocktails A and B. These cocktails and reasons for using them are not described in the main text and the table 4.

I have big problems with figure 6 and results displayed in it. 109 recordings were performed. 19 show TREK activity. What is with the remaining 90 recordings? How do the authors classify the nodose types with the single channel response? NPo as an abbreviation for open probability is new to me. Po is sufficient. In some cases/calculations N is the number of channels. The significance analysis in G-I is hard to believe. 3 stars in H for TREK1 and 1 for TREK2 and TRAAK. Maybe I am wrong, in that case sorry for that.

Overall, I like the idea to classify neurons with the help of different electrophysiological protocols, but support with other techniques like immunohistochemistry or  RT PCR would strongly support the story as well as measurement in heterologous expressions with the specific channels as a control.    

Author Response

Comment 1: Temperature sensing is important in physiology to control body function. The manuscript “Increasing temperature activates TREK potassium currents in vagal afferent neurons from the nodose ganglion” submitted by Rueda-Ruzafa et. al. classified three nodose ganglia profiles in response to temperature shifts, voltage protocols as well as modulation by specific compounds. It is important and interesting to look next to the well described TRP channels at other channels. However, I have some problems to follow the story. Often important aspects are not or to late introduced. For example line 42 – “All three NG neuronal types…,” The three types are introduced in line 44. Another example is the used cocktails A and B. These cocktails and reasons for using them are not described in the main text and the table 4.

Response 1: Thank you very much to the reviewer for the valuable comments. We apologize for the difficulties in following the manuscript and have revised several sections and figures to improve clarity and overall comprehensibility of the manuscript. You can find a new revised version of the manuscript uploaded.

Comment 2: I have big problems with figure 6 and results displayed in it. 109 recordings were performed. 19 show TREK activity. What is with the remaining 90 recordings? How do the authors classify the nodose types with the single channel response? NPo as an abbreviation for open probability is new to me. Po is sufficient. In some cases/calculations N is the number of channels. The significance analysis in G-I is hard to believe. 3 stars in H for TREK1 and 1 for TREK2 and TRAAK. Maybe I am wrong, in that case sorry for that.

Response 2: We apreciate the reviewer comments. Regarding single channel recordings, we apologize for the mistake. We patched 109 channels and 19 were TREK1 channels, 3 were TREK2 channels and 5 were TRAAK channels. We have now modified the word “recorded” by patched (line 245). Regarding nodose type classification in single channel recordings, the pink labels are a mistake and refer to the channel subfamily type (TREK1, TREK2 and TRAAK), not to NG neuronal types. We have corrected the pink labels. Also, we have change the abbreviation NPo for Po. Regarding the analysis, we thank the reviewer for the valuable comments and we have revised the analysis and reanalyze the data (paired sample t-test) and have checked that for TREK1 there is no statistical difference  in amplitude, but there is difference in Po (p<9,60879E-5) and for dwell time (p<9,93506E-4). Regarding TREK2, no difference was found in amplitude, but a p<0,01896 had resulted for Po and a p<0,03079 for dwell time. Last, TRAAK presented a difference in amplitude (p<0,03849), Po (p<0,01404) and dwell time (p<0,00816).

Comment 3: Overall, I like the idea to classify neurons with the help of different electrophysiological protocols, but support with other techniques like immunohistochemistry or RT PCR would strongly support the story as well as measurement in heterologous expressions with the specific channels as a control.    

Response 3: Thank you very much. Apart from the electrophysiological approach, we studied the expression of all the K2P family members (RT-qPCR) in mouse nodose ganglion and observed that TREK1 is the most expressed channel after TRESK in this ganglion. Part of this work has been already published (https://doi.org/10.1007/s12031-012-9780-y) comparing the obtained data of channel expression with those obtained from an autonomic motor ganglion (the superior cervical ganglion). If this reviewer thinks that adding this new information will ameliorate this work we can include a new figure reporting the relative expression of K2P channels (relative to housekeeping gene GADPH) in mouse nodose ganglion.

Round 2

Reviewer 1 Report

Comments and Suggestions for Authors

The authors have addressed the suggested comments, and I have no additional comments. I believe the revised manuscript contributes positively to the quality of the research.

Author Response

Thank you very much for your valuable comments. We really appreciate your review, and we are aware that this paper has thoroughly improved